# An Energy-Efficiency Prediction Method in Crude Distillation Process Based on Long Short-Term Memory Network

Yu Zhang [1], Zhe Cui [1,*], Mingzhang Wang [2], Bin Liu [1], Xiaomin Fan [1] and Wende Tian [1]

[1] College of Chemical Engineering, Qingdao University of Science & Technology, Qingdao 266042, China; tianwd@qust.edu.cn (W.T.)
[2] Sinopec Qingdao Petrochemical Co., Ltd., Qingdao 266043, China; wangmz.qdsh@sinopec.com
* Correspondence: cuiz@qust.edu.cn

**Abstract:** The petrochemical industry is a pillar industry for the development of the national economy affecting people's daily living standards. Crude distillation process is the core and leading unit of the petrochemical industry. Its energy consumption accounts for more than 20% of the total energy consumption of the whole plant, which is the highest energy consumption link. A model based on the long short-term memory network (LSTM) is proposed in this paper to predict and analyze energy efficiency. This model extracts the complex relationship between many process variables and predicts the energy efficiency of the crude distillation process. Firstly, the process simulation of crude distillation is carried out. By adding random disturbance, the data set in different working conditions is obtained, and the difference between the working conditions is expressed with the distance-coded heat map. Secondly, the Savitzky–Golay (SG) filter is used to smooth the data, which preserves the original characteristics of the data and improves the prediction effect. Finally, the LSTM model is used to predict and analyze the energy efficiency of products under different working conditions. The MAE, MSE, and MAPE results of the LSTM model under different working conditions in the test set are lower than 1.3872%, 0.0307%, and 0.2555%, respectively. Therefore, the LSTM model can be considered a perfect model for the test set, and the prediction results have high reliability to accurately predict the energy efficiency of the crude distillation process.

**Keywords:** energy consumption prediction; long short-term memory; crude distillation process; energy efficiency





## 1. Introduction

The petrochemical industry is the embodiment of comprehensive national strength and plays an important role in social development. The process of crude distillation is the link with the highest energy consumption in chemical plants and is the core and leading unit of the oil refining industry. This process is the first stage of crude oil processing to provide raw materials for the subsequent secondary processing. At present, the crude oil distillation process is more efficient and productive but still has the problem of high energy consumption. Based on the proposed national energy-saving policy and the economic costs associated with high energy consumption, the question of how to reduce energy consumption while maintaining production has become a top priority for factories. However, energy consumption cannot be predicted and analyzed over real time, resulting in operators being unable to monitor the change trend in energy efficiency timely.

The energy consumption of crude distillation is affected by numerous factors, such as the nature of crude oil, plant load, equipment operation, and production cycle. The reduction of energy consumption can improve the economic benefit of enterprises and make rational use of petroleum resources. Common energy saving and consumption reduction are carried out on the basis of existing equipment conditions. By changing some control parameters, such as feed temperature, feed flow rate, tower bottom temperature, and reflux

ratio, the process operation conditions can be optimized to reduce energy consumption. Optimizing the energy integration of the crude distillation system and heat exchanger system can reduce energy consumption [1], the optimization of operating conditions can improve economic benefits [2], and a genetic algorithm is used to improve the production to achieve the balance between profit and energy consumption by multi-objective optimization problems [3]. Yang et al. [4] proposed an optimal operation strategy to improve the energy efficiency of a crude oil distillation unit without any structural transformation. The detailed process operation was expressed as a complex nonlinear programming model and then solved with the double-loop algorithm. It can help engineers easily adjust the key parameters of the process. Yao et al. [5] optimized the operational variables of the simulated atmospheric distillation column by designing experimental techniques and support vector regression models. Li et al. [6] introduced a knowledge-based operational optimization strategy to mitigate uncertainties in the properties of the materials. It combines neural network and fuzzy logic technology to provide instructions to adapt to different material properties. Ochoa Estopier et al. [7] proposed a framework for thermally integrated crude oil distillation systems and developed an artificial neural network model and a heat exchanger network modification model. The above methods cannot achieve the prediction of energy efficiency and need to establish a complex mechanism model. Deep learning is a novel prediction method that can accurately predict energy efficiency.

With the advent of the era of big data, deep learning and artificial neural networks have been increasingly used in the chemical industry with their self-learning, self-adaptive, and self-organizing characteristics. It has improved the identification and diagnosis ability of abnormal energy consumption of chemical enterprises with a new solution for the energy saving of chemical companies [8]. Accurate prediction of energy consumption of crude distillation is the basis of energy management and control and can be used by managers to optimize decisions [9,10]. Energy-efficiency prediction models can promote the efficient use of energy and low consumption of raw materials. Convolutional neural networks (CNNs) are the most effective deep learning networks for modeling complex processes. Qi et al. [11] proposed a multi-operation mode-adaptive time-window convolutional neural network (MOM-ATWCNN) for energy consumption prediction, which shows its superiority in various performance indicators. The improvement of the algorithm is beneficial to reducing energy consumption and achieving economic goals. Navid Fekri et al. [12] proposed an online adaptive recurrent neural networks (RNNs) model for power load prediction. By emphasizing newly arrived data and adaptive load changes, the prediction accuracy is improved and superior to several other models. Zhang et al. [13] proved the effectiveness of CNNs in the power generation prediction. Although the key features are affected by multiple factors, the method can express these features more accurately and completely.

The data of the crude distillation process are characterized by multi-dimension and uncertainty. Energy consumption prediction for crude distillation is helpful in solving the problem of energy consumption target setting and scheduling optimization. The predicted results and optimized values are of great significance for reducing energy consumption, guiding crude oil production, and improving energy efficiency [14]. RNNs have problems such as gradient disappearance or gradient explosion and can only learn short-term influence relationships. The long short-term memory network (LSTM), as a variant of the cyclic neural network, can effectively solve the above problems. In addition, much time-domain correlation data exist in the field of the chemical industry, so LSTM has great advantages for prediction. Han et al. [15] proposed a fault diagnosis method for chemical processes based on optimized LSTM to determine the optimal number of hidden layer nodes under different fault conditions and improve the accuracy of fault diagnosis. Xu et al. [16] proposed a prediction method for pipeline leakage of heat exchangers based on generative adversarial networks (GANs) and LSTM. The data enhancement method of GAN is used to solve the problem of data imbalance, a classifier based on LSTM is used to solve the problem of time dependence of process data, and the pipeline state is classified to predict leakage. Han et al. [17] proposed an attentional mechanism-based production capacity analysis and

energy-saving model of LSTM, established a production prediction model with LSTM, and applied the model to predict the production capacity, providing theoretical guidance for improving production capacity. Zhu et al. [18] used LSTM to predict the operating profit of the natural gas–liquid recovery device so as to optimize the operating conditions of the process and improve the profit rate of the device.

In this paper, an energy-efficiency prediction method that combines mechanistic modeling and artificial intelligence is proposed. There is a close relationship between the energy efficiency of the crude distillation and the process parameters, and the real-time changes in parameters affect the energy-efficiency level of the product. By adding disturbance, the data of four typical operating conditions of the crude distillation units are simulated, and the distance-coded heat map is introduced to realize the visualization of the differences in operating conditions. The Savitzky–Golay filter is used to smooth the data and establish the sample set. The trained LSTM model is used to predict the energy efficiency of atmospheric two-line products and vacuum two-line products. The predicted energy-efficiency value can serve as a guide in the production process. Based on the input operating parameters, the LSTM model outputs predicted energy-efficiency values that can guide operators in making decisions to optimize operating parameters, energy-efficiency diagnostics, and applying optimal production strategies to improve energy-efficiency.

The rest of this paper is organized as follows. The second section introduces the proposed method and its theoretical basis. The third section introduces the process simulation of crude distillation. The fourth part predicts and analyzes the energy efficiency of the crude distillation. The last part summarizes the work of this paper.

## 2. Methodology and Theoretical Basis

### 2.1. Proposed Method

As shown in Figure 1, the method consists of three parts: (1) construction of mechanism model of crude distillation process; (2) process data processing; and (3) energy-efficiency prediction based on LSTM.

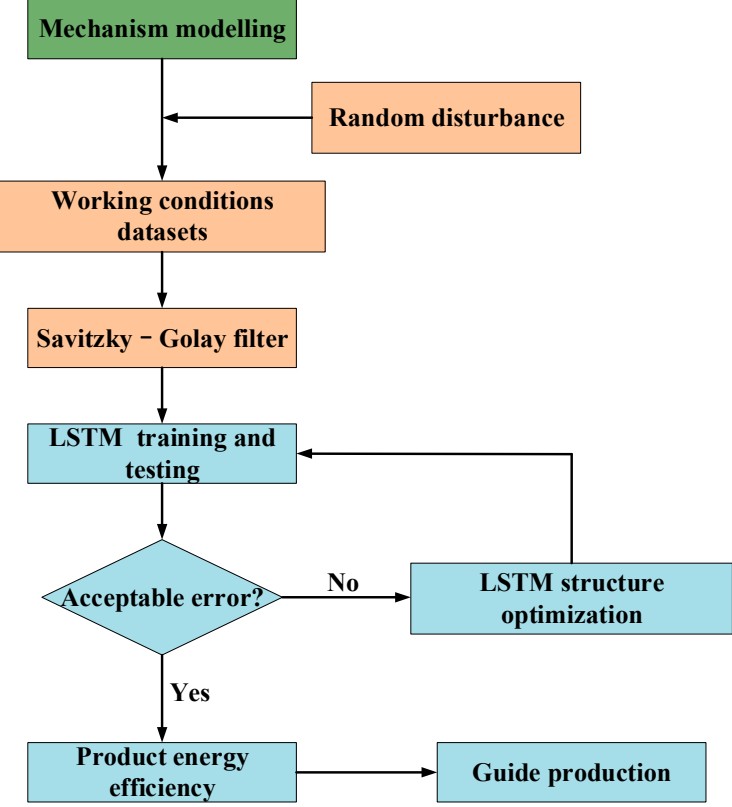

**Figure 1.** The framework of proposed method.

In the part of mechanism modeling, based on the software characteristics of Aspen HysysV10, the steady-state model is first established to obtain the dynamic model. The ideal value obtained by steady-state simulation is used as a reference to evaluate the dynamic fluctuation degree of parameters.

In the data processing part, real-time changes in parameters affect the energy efficiency of the product. By adding random disturbance to the operating parameters, the data sets under four operating conditions are obtained, and the differences between operating conditions are demonstrated by using distance-coded heat maps. Savitzky–Golay filtering (SG) is used to reduce the noise of data and improve the prediction effect while preserving the original characteristics of data.

In the part of energy-efficiency prediction, the data after noise reduction are divided into a train set and a test set in a ratio of 8:2. The training set is used to train the model, and the test set is used to predict the energy efficiency. Under different working conditions, the LSTM model can predict accurately.

### 2.2. Process Simulation

Aspen Hysys has comprehensive thermodynamic data libraries for energy optimization and cost estimation. Compared with Aspen Plus and Aspen Dynamics, it allows for the conversion between steady-state and dynamic simulations in the program instead of switching between different programs [19,20]. By using Aspen Hysys to build the mechanism model, accurate simulation results can be obtained, and engineering efficiency can be improved [21]. Therefore, the mechanism model used in this study is obtained through Aspen Hysys, and simulation results are then obtained to analyze and improve energy efficiency.

The steady-state model is built by using the loading data of the chemical process. The simulation results of the steady-state simulation are consistent with the actual production state. However, the actual chemical process is in an unstable state, and the problems in the process can not be solved by the steady-state simulation method. After building a reasonable steady-state model, dynamic import is carried out. The real operating state of the device is simulated, and the real-time data of the operating parameters are collected.

### 2.3. LSTM Model

Energy-efficiency prediction and analysis work is based on in-depth analysis of historical data, which are collected and analyzed. Artificial intelligence algorithm is used to learn the rules and internal information of the data. Compared with traditional data processing algorithm methods, the LSTM model has a better prediction effect on the time-series data. As a result, the prediction results with high accuracy will help people to make efficient decisions in time.

Recurrent neural networks (RNNs) are self-connected neural networks in the field of deep learning. Using neurons with self-feedback functions, it can process time sequence data of any length. RNNs with multiple hidden layers are composed of an input unit, output unit, and hidden unit, as shown in Figure 2. $x_t$ is the input at time t, and the input set is marked as $\{x_0, x_1, \ldots, x_t, x_{t+1}, \ldots\}$; $o_t$ is the output at time t, and the output set is labeled $\{y_0, y_1, \ldots, y_t, y_{t+1}, \ldots\}$; $S_t$ is the hidden state at time t, and the hidden unit is marked as $\{S_0, S_1, \ldots, S_t, S_{t+1}, \ldots\}$. The above can be expressed as:

$$S_t = f\left(Ux_t + Ws_t - 1\right) \tag{1}$$

where f is a nonlinear activation function, such as tanh, sigmoid, and ReLU.

RNNs can achieve short-term memory of time-series data. When the output information is close to node information, the model can make suitable use of historical information. However, it is difficult for the RNN model to make full use of the effective information for the long-time node case. In addition, RNNs have problems such as gradient vanishing or gradient explosion, which makes it impossible to learn the long-term influence relationship [22]. LSTM, as one of the most successful variants of recurrent neural networks, can

effectively solve the problem of continuous data input without preservation [17]. The cell structure is shown in Figure 3.

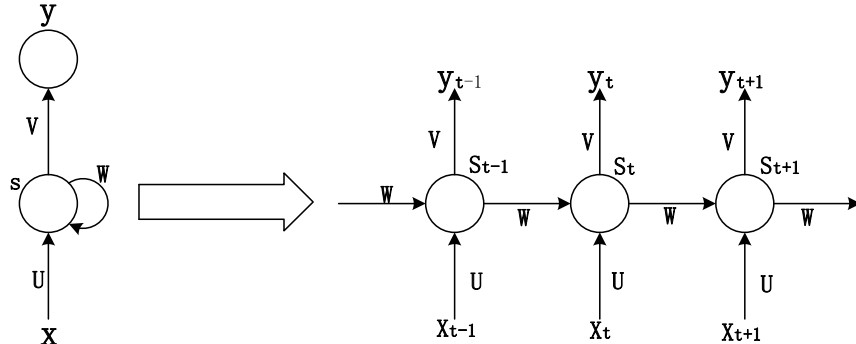

**Figure 2.** Recurrent neural network model with multiple hidden layers.

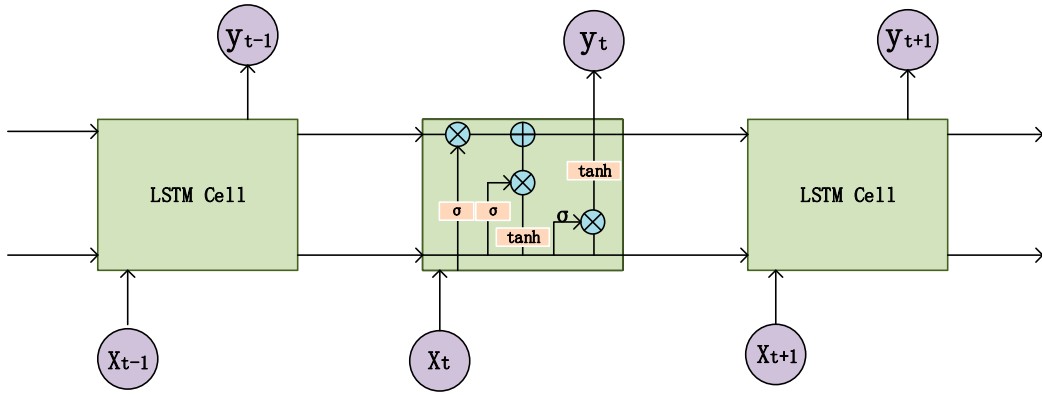

**Figure 3.** LSTM internal unit structure.

Compared to RNNs, LSTM has a forgetting gate, input gate, and output gate. The input gate controls the amount of information flowing into the storage unit; the forgetting gate controls the proportion of information accumulated in the unit from the previous moment to the current moment; and the output gate controls the proportion of hidden state information [23]. LSTM has the advantage of adding a forgetting mechanism as follows. When there are new input samples, the model will judge which historical information needs to be deleted. When the model inputs new samples, it will automatically determine whether to use and save the features and control the transfer state by the gating mechanism to maintain a certain gradient, then show suitable performance. The calculation formulas are as follows.

$$i_f^{(t)(v\tau)} = \sigma\left(\sum_{f'=0}^{F_{v-1}-1} \Theta_{f'}^{i_v(v)f} h_{f'}^{(t)(v-1\tau)} + \sum_{f'=0}^{F_v-1} \Theta_{f'}^{i_\tau(v)f} h_{f'}^{(t)(v\tau-1)}\right) \tag{2}$$

$$f_f^{(t)(v\tau)} = \sigma\left(\sum_{f'=0}^{F_{v-1}-1} \Theta_{f'}^{f_v(v)f} h_{f'}^{(t)(v-1\tau)} + \sum_{f'=0}^{F_v-1} \Theta_{f'}^{f_\tau(v)f} h_{f'}^{(t)(v\tau-1)}\right) \tag{3}$$

$$o_f^{(t)(v\tau)} = \sigma\left(\sum_{f'=0}^{F_{v-1}-1} \Theta_{f'}^{o_v(v)f} h_{f'}^{(t)(v-1\tau)} + \sum_{f'=0}^{F_v-1} \Theta_{f'}^{o_\tau(v)f} h_{f'}^{(t)(v\tau-1)}\right) \tag{4}$$

where $i_f^{(t)(v\tau)}$ is the input gate of the LSTM unit, $g_f^{(t)(v\tau)}$ is the forgetting gate of the LSTM unit, $o_f^{(t)(v\tau)}$ is the output gate of the LSTM unit, and $\sigma$ is the activation function.

### 2.4. Savitzky–Golay Filter

Savitzky and Golay proposed the Savitzky–Golay filter in 1964, which is a filter method that can be fitted in the time domain based on the local polynomial least square method [24].

This method can remove noise while keeping the width and shape of the signal unchanged. When the SG filter reduces noise at a point, it needs to fit and calculate the surrounding points so there are not enough data points when processing the first and last data. In order to overcome the phenomenon of suppressing high frequency and artifacts, we remove the front-end and back-end data points and keep the data that have enough data points to fit.

The variables of crude distillation have the characteristics of high coupling and non-linearity, which are easily affected by operating parameters and the external environment. When random disturbance is added, there will be noise in the collected data center. Noise will affect the prediction accuracy of the LSTM model, so data noise reduction is a prerequisite for the accurate prediction of energy efficiency [25]. The SG filter can directly smooth the data from the time domain without the traditional filter between the frequency domain and time conversion, so the filter is widely used for data smoothing and denoising [24].

### 2.5. Distance-Coded Heat Map

Euclidean distance [26,27] is an intuitive distance measurement method used to measure the absolute distance between two points in space. The formula is shown in Equation (5).

$$d = \sum_{k=1}^{n} \sqrt{(x_1 - x_2)^2} \tag{5}$$

Euclidean distance can be used to determine the degree of similarity between data. The smaller the calculated value, the higher the degree of similarity between individuals. This paper introduces the method to calculate the difference of operating parameters in different working conditions. The similarity between operating parameters can be reflected according to the Euclidean distance value.

A heat map is a matrix that reflects data through color changes, which can show the correlation between different indicators and different data. Data can be visually displayed through the heat map, enhancing readability and visualization. In this paper, a distance-coded heat map is introduced to calculate the values between different working conditions by using Euclidean distance, and the differences between the data of different working conditions are reflected in the way of a heat map.

## 3. Process Simulation of Crude Distillation

### 3.1. Process Description

Crude oil is mainly composed of C, H, S, N, O, and other elements. In addition, there are trace metal elements and other non-metallic elements. In refineries, crude oil is usually cut into several fractions and evaluated using crude distillation curves. A true boiling point distillation curve (TBP) can well represent the relationship between oil temperature and composition. Under the condition of a mass reflux ratio of 5:1, a separation distillation column with a theoretical plate number of 14–18 was used to separate the light and heavy fractions. A temperature interval of 10 °C or a mass fraction of 3% is generally used as a narrow fraction to calculate the total yield or the yield per component. The real boiling point distillation curve of crude oil is shown in Figure 4, where the vertical axis is the percentage of distillate, and the horizontal axis is the distillation temperature. This curve can reflect the true boiling point of each component in the distillate.

The crude distillation device is mainly composed of an electric desalting device, pre-flash column, heating furnace, atmospheric column, vacuum column, and so on. After the flash treatment, the flash top gas enters the atmospheric column, and the flash bottom oil enters the atmospheric tower after being heated by the atmospheric furnace. The process of vaporizing, separating, and cooling crude oil runs under atmospheric operating conditions. In order to effectively utilize the heat of the steam and regulate the gas phase load in the tower, three steam lift towers and three mid-pumparound refluxes are set up.

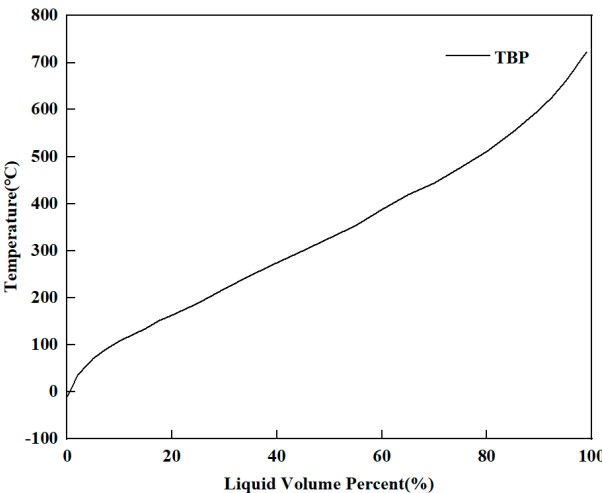

**Figure 4.** True boiling point distillation curve of crude oil.

In atmospheric distillation, product fractions with different boiling points can be separated by controlling the temperature at the top of the tower, the extraction temperature at each sideline, the amount of stripping steam, and the flow of reflux in each middle section. Kerosene and diesel oil are obtained on the sideline, and fuel gas is obtained on the top of the tower. The residue obtained at the bottom of the tower is heated by a vacuum furnace and then enters the vacuum column.

If the heavy oil is distilled at atmospheric pressure, the colloid, asphaltene, and some unstable groups in the heavy oil will trigger cracking and a condensation reaction, resulting in product quality reduction and cooking equipment. The components simulated in this study are mostly organic compounds, and the Antoine equation is usually used to calculate the relationship between the temperature and vapor pressure of organic compounds. As shown in Equation (6), the Antoine equation considers the relationship between vapor pressure and molecular weight, chemical structure, system temperature, and other factors. Compared with other complex equations, the calculation accuracy of the Antoine equation is accurate with simple steps.

$$\ln P = A - B/(T + C) \tag{6}$$

where P is the vapor pressure of the component, T is the system temperature, and A, B, C is the physical property constant of the component.

In order to further treat atmospheric residual oil, it is necessary to reduce the external pressure to reduce the boiling point of the substance. The vacuum column is set up by three middle refluxes, and the bottom of the tower concentrates most of the gum, asphaltene, and a very high boiling point of the oil. The simulation diagram of the crude distillation device is shown in Figure 5.

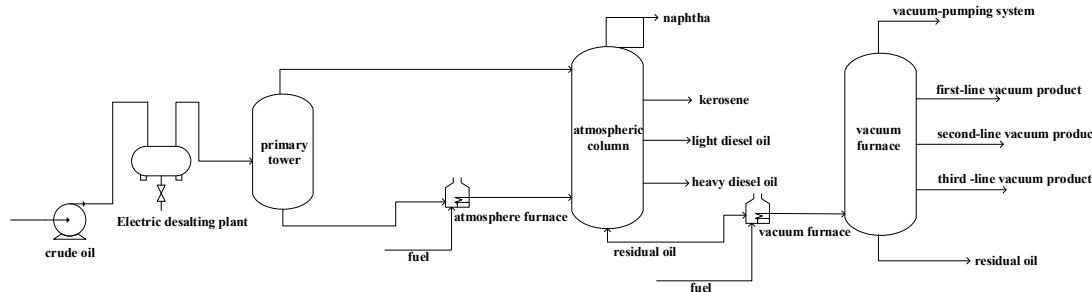

**Figure 5.** Flow chart of crude distillation.

### 3.2. Crude Distillation Simulation Parameter

Aspen Hysys is used to establish the steady-state model of crude distillation. The parameters of the model are state parameters of the chemical process. MAE is introduced to reflect the fluctuation range of dynamic values to prove the representativeness and reliability of steady-state operating parameters. As a result, the dynamic behaviors of parameters fluctuate around the steady-state value, and the MAE value is less than 0.5. The parameters of the steady-state simulation are set as follows.

In the process simulation, the atmospheric column adopts a plat tower, the plate number is 50, the top temperature is 142 °C, the top pressure is 150 kPa, and the total tower pressure drop is 125 kPa. A condenser is set at the top of the tower, and its temperature is set at 71 °C. The temperature and flow parameters of lateral oil are shown in Table 1.

**Table 1.** The basic parameters of side line products of the atmospheric column.

| Oil Product | Name of Parameter | Data |
|---|---|---|
| Naphtha | Temperature °C | 71.26 |
| | Flow t/h | 90 |
| One-line products | Temperature °C | 181.2 |
| | Flow t/h | 20 |
| Two-line products | Temperature °C | 174.4 |
| | Flow t/h | 20 |
| Three-line products | Temperature °C | 195.9 |
| | Flow t/h | 20 |
| Residue | Temperature °C | 354.1 |
| | Flow t/h | 454.8 |

In the atmospheric tower, we set up three mid-pumparounds. The circulation reflux takes heat from the high temperature and returns it to the tower from the upper part. The relevant parameters of mid-pumparounds are shown in Table 2.

**Table 2.** Circulating parameters of the atmospheric column in the middle section.

| Mid-Pumparound | Extraction Plate Position | Return Plate Position | Flow t/h | Heat Duty kJ/h |
|---|---|---|---|---|
| One-line mid-pumparound | 4 | 1 | 200 | $4.362 \times 10^7$ |
| Two-line mid-pumparound | 20 | 17 | 142.3 | $3.014 \times 10^7$ |
| Three-line mid-pumparound | 32 | 28 | 357.2 | $3.572 \times 10^7$ |

As shown in Table 3, the feed temperature of the atmospheric column is 350 °C, the feed flow rate is 568.8 t/h, and the feed pressure is 267.3 kPa. The temperature at the top of the tower is 142 °C, and the temperature at the bottom is 357 °C. The top pressure is 150 kPa, and the tower pressure drop is 125 kPa.

**Table 3.** Basic parameters of the atmospheric column.

| Parameter | Value |
|---|---|
| Feed temperature °C | 350 |
| Feed flow t/h | 568.8 |
| Feed pressure kPa | 267.3 |
| Top temperature °C | 142 |
| Bottom temperature °C | 357 |
| Top pressure kPa | 150 |
| Bottom pressure kPa | 275 |

The atmospheric residual oil enters the vacuum column after being heated by the heating furnace. The number of vacuum column plates is 44 layers. The temperature of the tower top is 209.8 °C, the pressure of the tower top is 4 kPa, and the pressure drop of the whole tower is 25.61 kPa. Vapor extraction steam is set at the bottom of the tower to reduce the partial pressure of oil and gas and to improve the gasification rate of the feed material. The temperature and flow parameters of lateral oil are shown in Table 4.

**Table 4.** The basic parameters of sideline products of the vacuum column.

| Oil Product | Name of Parameter | Data |
|---|---|---|
| One-line products | Temperature °C | 238.9 |
| | Flow t/h | 18.98 |
| Two-line products | Temperature °C | 263.5 |
| | Flow t/h | 18.76 |
| Three-line products | Temperature °C | 190 |
| | Flow t/h | 20.12 |
| Four-line products | Temperature °C | 284.5 |
| | Flow t/h | 18.82 |
| Residue | Temperature °C | 373.6 |
| | Flow t/h | 264.4 |

In the vacuum tower, we also take the way of mid-pumparounds to take the heat, the gas–liquid load in the tower is evenly distributed, and energy is saved. The relevant parameters of mid-pumparounds are shown in Table 5.

**Table 5.** Circulating parameters of vacuum column in the middle section.

| Mid-Pumparound | Extraction Plate Position | Return Plate Position | Flow t/h | Heat Duty kJ/h KJ/h |
|---|---|---|---|---|
| One-line mid-pumparound | 10 | 1 | 74.63 | $1.52 \times 10^7$ |
| Two-line mid-pumparound | 20 | 11 | 131.1 | $2.347 \times 10^7$ |
| Three-line mid-pumparound | 30 | 21 | 72.19 | $1.59 \times 10^7$ |

As shown in Table 6, the feed temperature of the vacuum tower is 326.2 °C, the feed flow rate is 454.8 t/h, and the feed pressure is 22.52 kPa. The temperature at the top of the tower is 209.8 °C, and the temperature at the bottom is 382.1 °C. The top pressure is 4 kPa, and the tower pressure drop is 25.61 kPa.

**Table 6.** Basic parameters of vacuum column.

| Parameter | Value |
|---|---|
| Feed temperature °C | 326.2 |
| Feed flow t/h | 454.8 |
| Feed pressure kPa | 22.52 |
| Top temperature °C | 209.8 |
| Bottom temperature °C | 382.1 |
| Top pressure kPa | 4 |
| Bottom pressure kPa | 29.61 |

Steady-state processes are generally temporary and relative. The actual production process is always subject to various fluctuations, disturbances, and changes in conditions. Therefore, after building a steady-state model, controllers and control loops are added to realize the dynamic simulation of the device.

### 3.3. Energy-Efficiency Analysis

The energy consumption of crude distillation is affected by many factors, such as the nature of crude oil, plant load, equipment operation, and production cycle. Predicting and improving operating conditions is also an important way to save energy and reduce consumption. At present, the energy saving of crude distillation is mainly focused on reducing the consumption of fuel gas, electricity, steam, and water. Fuel consumption accounts for a large proportion of the total energy consumption, mainly consumed in the heating furnace, so it is very necessary to analyze the fuel consumption of equipment. Stripping steam is needed in the process of crude distillation, and stripping steam is mainly used at the bottom of atmospheric and vacuum towers, as well as stripping towers. The cost of stripping steam is high, and the demand is great. Therefore, it is necessary to monitor the steam pressure at each site and analyze the relevant energy consumption. The main consumption of water is interrupted by circulation water and condenser water. The main power consumption equipment is the machine pump, which is responsible for the transportation and compression of materials.

It is necessary to establish a set of indexes to evaluate the energy efficiency according to the energy consumption characteristics of crude distillation in order to guide enterprises to improve energy-saving schemes and improve the level of energy efficiency. Energy consumption is the sum of fuel energy consumption, steam energy consumption, cooling water energy consumption, and electricity energy consumption. Energy efficiency is a comprehensive index that combines production input and output [28]. Its definition is as follows:

$$\text{Energy efficiency} = \frac{\text{energy consumption}}{\text{product output}} \tag{7}$$

where product output is the production of crude distillation, and the unit is kg/h. Energy consumption is the sum of all the energy consumed, and the unit is kW. By definition, the higher the level of energy efficiency, the better the enterprise's energy use and the greater the economic benefits.

The operating parameters of the process are characterized by high coupling and nonlinearity. Different operating parameters will affect the energy efficiency of the unit. The fluctuation of energy consumption and output is caused by the change in operating conditions, such as the amount of crude oil feed, temperature, and steam at the bottom of the tower during production. The single working condition is not able to reasonably evaluate the energy-efficiency level and better train the LSTM energy-efficiency prediction model. Therefore, when predicting and analyzing the energy efficiency of crude distillation, the division of working conditions is required.

## 4. Energy Consumption Prediction and Analysis

### 4.1. Mathematical Case

In mathematical cases, the relationship between five input variables, X1, X2, X3, X4, and X5, and one output variable, Y, is simulated. The relationship between Y and X is expressed as:

$$X1 = 3\sin t + 1 \tag{8}$$

$$X2 = 6\cos t + 3 \tag{9}$$

$$X3 = 3\sin 2t \tag{10}$$

$$X4 = 4\sin 3t + 5 \tag{11}$$

$$X5 = \cos 9t \tag{12}$$

$$Y = 2X1 + 3X2 + 4X3 - X4 + X5 \tag{13}$$

where t is time, X is the input variable of the model, and Y is the output variable of the model.

A total of 1000 time sequence data are generated, and the ratio of the training set to the test set is 8:2, namely 800 training sets and 200 test sets. LSTM is trained, and the Y value can be predicted according to the relationship between X and Y. The predicted results are shown in Figure 6.

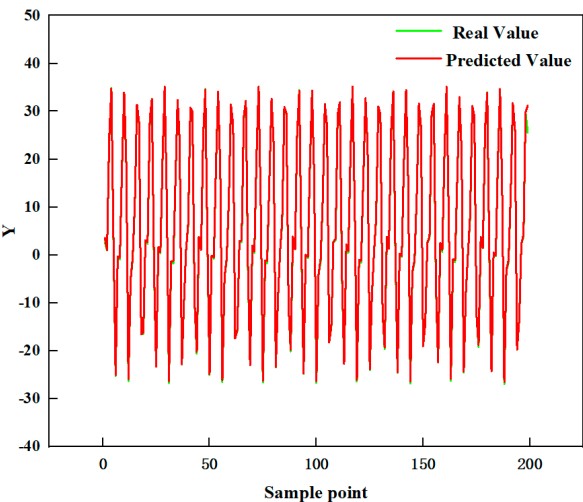

**Figure 6.** The prediction of Y value.

In Figure 6, the horizontal axis is the sample point, and the vertical axis is the Y value. The green line shows the true Y value, and the red line shows the predicted Y value. It can be seen that the red line and green line have a suitable fitting degree, and the model has a high accuracy and suitable applicability in forecasting.

*4.2. Data Preprocessing*

In order to better train the LSTM model for energy-efficient prediction, disturbances are added randomly, and four working conditions are divided. The energy efficiency of the atmospheric two-line products and vacuum two-line products are predicted for each working condition. Each set of working conditions consists of 2400 time-series data, where the ratio of the training set and test set is divided according to 8:2; that is, 1920 for the training set and 480 for the test set. The operational variables of the four operating conditions data are used as inputs to the LSTM model, and the energy efficiency of the product is used as the output. The randomly added disturbances are visualized using distance-coded heat maps that show the differences between operating conditions.

Energy efficiency is closely related to operating parameters. Real-time changes in parameters can have an impact on energy efficiency. Thirteen operational parameters related to the energy efficiency of products are selected, disturbances are added, and data sets are generated to predict the energy efficiency of the atmospheric two-line products. We select 11 operational parameters related to the energy-efficiency products, add disturbances, and generate a data set to predict the energy efficiency of vacuum atmospheric two-line products.

The influence of selected operating parameters on the energy-efficiency level is expounded. The outlet temperature of the heating furnace is related to the mass flow rate of the fuel, and the increase in outlet temperature can improve the gas rate of the crude oil, thus increasing the quality of the sideline product. Top pressure and temperature will also affect the vaporization rate of oil. The lower the top pressure is, the boiling point will decrease correspondingly, and the vaporization rate in the tower will increase. Stripping

steam can reduce the partial pressure of oil and gas in the tower, reduce the boiling point of components, and increase the production of sideline products. Mid-pumparound can recover the heat at high temperatures, increase the processing capacity, and then affect energy consumption. Tables 7 and 8 show the selected operating parameters. Tables 9 and 10 show the dataset size.

**Table 7.** Prediction parameters of atmospheric two-line energy efficiency.

| Variable No. | Variable Description | Variable No. | Variable Description |
|---|---|---|---|
| 1 | Crude oil mass flow | 8 | Amount of condensate |
| 2 | Temperature of crude oil | 9 | One-line mid-pumparound |
| 3 | Tower bottom steam flow | 10 | Two-line mid-pumparound |
| 4 | One-line steam mass flow | 11 | Three-line mid-pumparound |
| 5 | Three-line steam mass flow | 12 | Top pressure |
| 6 | Fuel flow | 13 | Top temperature |
| 7 | Furnace outlet temperature | | |

**Table 8.** Prediction parameters of vacuum two-line energy efficiency.

| Variable No. | Variable Description | Variable No. | Variable Description |
|---|---|---|---|
| 1 | Crude oil mass flow | 7 | One-line mid-pumparound |
| 2 | Temperature of crude oil | 8 | Two-line mid-pumparound |
| 3 | Tower bottom steam flow | 9 | Three-line mid-pumparound |
| 4 | Fuel flow | 10 | Top pressure |
| 5 | Fuel flow | 11 | Top temperature |
| 6 | Amount of condensate | | |

**Table 9.** Specification of atmospheric two-line data sets.

| Samples | | Attributes | |
|---|---|---|---|
| Training | Testing | Inputs | Output |
| 1920 | 480 | 13 | 1 |

**Table 10.** Specification of vacuum two-line data sets.

| Samples | | Attributes | |
|---|---|---|---|
| Training | Testing | Inputs | Output |
| 1920 | 480 | 11 | 1 |

In order to show the difference between normal conditions and randomly disturbed conditions, a sliding window was introduced to select the data of different conditions. The Euclidian distance formula was used to calculate the dynamic distance between conditions so as to generate the distance-coded heat map. As shown in Figures 7–10, the vertical axis of the heat map represents all parameters that affect product energy efficiency, and the horizontal axis represents the time step of the sliding window. The selected sliding window is 100, and sliding sampling is carried out between different data. A total of 2400 data are selected for each working condition, so the time step of the sliding window is 24. The axis on the right of the heat map represents the Euclidean distance between normal conditions and randomly disturbed conditions, and the depth of the color represents the degree of difference between conditions. The brighter the color, the greater the random disturbances added by the parameter. The darker the color, the less significant the disturbances of the working parameters. With the distance-coded heat map, the data differences between

working conditions can be displayed visually and clearly to visualize the differences in working conditions.

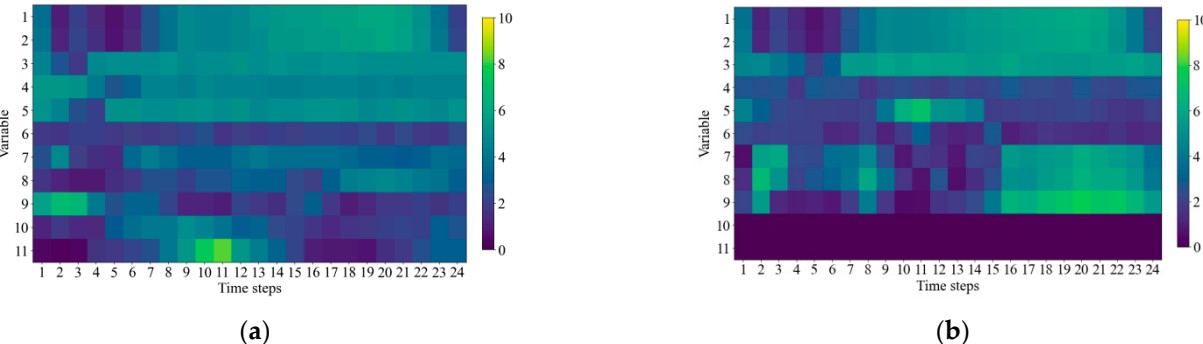

(**a**)                                (**b**)

**Figure 7.** Distance-coded heat map of working condition 1. (**a**) Heat map of operating parameters of atmospheric two-line products. (**b**) Heat map of operating parameters of vacuum two-line products.

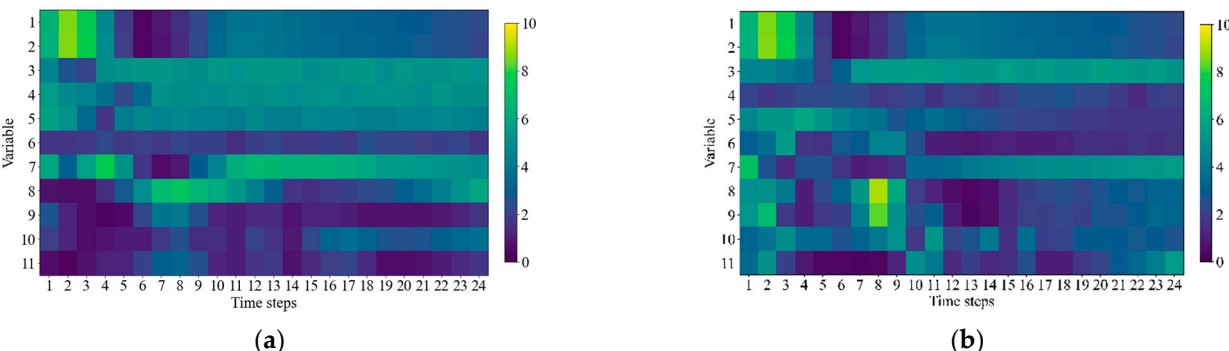

(**a**)                                (**b**)

**Figure 8.** Distance-coded heat map of working condition 2. (**a**) Heat map of operating parameters of atmospheric two-line products. (**b**) Heat map of operating parameters of vacuum two-line products.

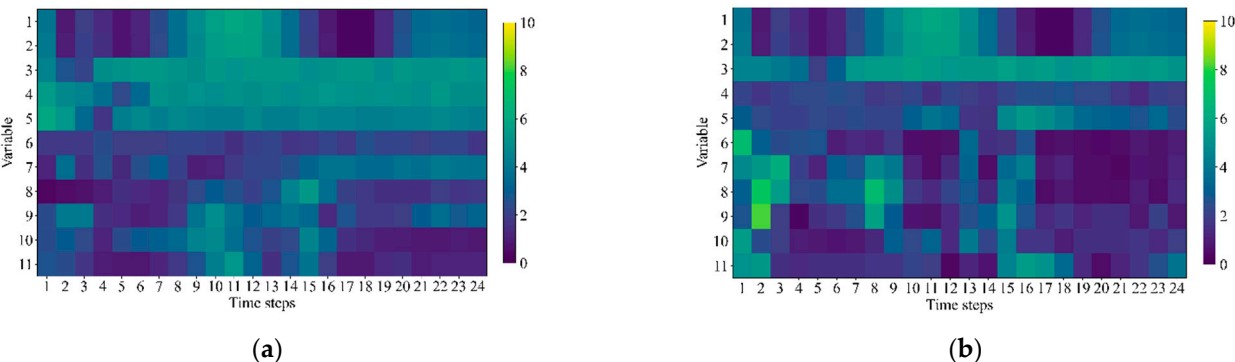

(**a**)                                (**b**)

**Figure 9.** Distance-coded heat map of working condition 3. (**a**) Heat map of operating parameters of atmospheric two-line products. (**b**) Heat map of operating parameters of vacuum two-line products.

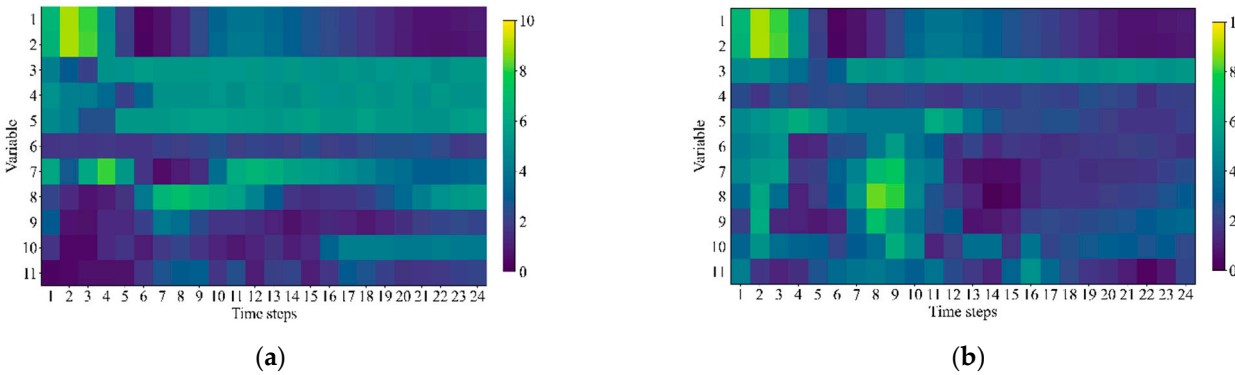

(**a**)                                    (**b**)

**Figure 10.** Distance-coded heat map of working condition 4. (**a**) Heat map of operating parameters of atmospheric two-line products. (**b**) Heat map of operating parameters of vacuum two-line products.

*4.3. LSTM Model Prediction*

4.3.1. Data Normalization and Evaluation Criterion

In data processing, different indicators have different dimensions and units. In order to ensure the accuracy and comparability of data analysis results, the data need to be normalized. After normalization processing, each index is in the same order of magnitude, and the result is between [0, 1]. In this case, a comprehensive evaluation of data can be carried out. Normalization can improve the accuracy and convergence speed of the model, and its calculation formula is as follows:

$$X^* = \frac{x - \min}{\max - \min} \tag{14}$$

where min is the minimum value of the data sample; max indicates the maximum value of the data sample.

As shown in Equation (15), mean absolute error (MAE) is the average absolute error between real values and predicted values. Moreover, the smaller value of MSE indicates the higher accuracy of the model prediction.

$$MAE = \frac{1}{n}\sum_{i=1}^{n}|y_i - \hat{y}_i| \tag{15}$$

where $y_i$ is the true value and $\hat{y}_i$ is predicted value.

Mean Square Error (MSE), as shown in Equation (16), is the average value of the squared sum of errors between real values and predicted values and can be explained as follows: the smaller value of MSE, the higher accuracy of the model prediction.

$$MSE = \frac{1}{n}\sum_{i=1}^{n}(y_i - \hat{y}_i)^2 \tag{16}$$

As shown in Equation (17), mean absolute percentage error (MAPE) is also calculated not only to represent the absolute error between real values and predicted values but also to characterize the error ratio to real values.

$$MAPE = \frac{1}{n}\sum_{i=1}^{n}\left|\frac{y_i - \hat{y}_i}{y_i}\right| \tag{17}$$

4.3.2. LSTM Hyperparameters Optimization

The orthogonal experimental method is an experimental method to study multiple factors and multiple levels based on the principles of orthogonality and homogeneity. Partial experiments instead of comprehensive experiments can accurately find the optimal parameter combination and optimize multiple hyperparameters at the same time. Compared with the non-orthogonal test, it has the advantages of high precision and high

efficiency. Hyperparameters affect the accuracy of the LSTM energy-efficiency prediction model. The orthogonal experiment method can optimize the hyperparameters and improve the accuracy of the model.

The predictive effect of LSTM is related to the activation function, batch size, and number of network layer nodes. The number of network layer nodes determines the generalization ability and learning ability of the network. However, too complex a network structure will bring about the overfitting problem, and the increase in the number of network layer nodes will also lead to an increase in training costs. Common activation functions are sigmoid, relu, and tanh. Different activation functions will affect the predictive performance of the model. The sigmoid is a smooth step function that can be used for binary classification. The tanh activation function outputs values between $[-1, 1]$, which mitigates the gradient disappearance problem. The relu activation function keeps only positive elements and zeroes out negative elements. Batch size is the number of samples in the model at each training, and its size affects the degree and speed of optimization of the model. A suitable number of batches can reduce the number of iterations during network training, improve the training speed, and make the gradient descent direction more accurate.

To improve the accuracy of the model prediction, three levels of three factors is set up to find the superiority of the hyperparameters. The parameters are set as a: batch size (50, 100, 150), b: activation function (relu, sigmoid, tanh), and c: number of network layer nodes (64, 32, 128). The details of the orthogonal test hyperparameters are listed in Table 11, and the predicted results are shown in Figure 11.

**Table 11.** The scheme of orthogonal experiment.

| Plan Number | Horizontal Combination | Optimization Parameter | | |
|---|---|---|---|---|
| | | Number of Network Layer Nodes | Activation Function | Batch Size |
| A | $a_1b_1c_1$ | 50 | relu | 64 |
| B | $a_1b_2c_2$ | 50 | sigmoid | 32 |
| C | $a_1b_3c_3$ | 50 | tanh | 128 |
| D | $a_2b_1c_2$ | 100 | relu | 32 |
| E | $a_2b_2c_3$ | 100 | sigmoid | 64 |
| F | $a_2b_3c_1$ | 100 | tanh | 128 |
| G | $a_3b_1c_3$ | 150 | relu | 128 |
| H | $a_3b_2c_1$ | 150 | sigmoid | 64 |
| I | $a_3b_3c_2$ | 150 | tanh | 32 |

In Figure 11, the horizontal axis represents the number of sample points in the test set, and the vertical axis represents the energy efficiency of the crude distillation process. Moreover, the black line represents the real energy efficiency, and the red line shows the energy efficiency predicted by the LSTM model. The figure can be explained below: the better the two lines fit, the more accurate the model prediction results. It can also be seen from Figure 10 that the prediction of the model differs with different hyperparameters. Plan (B) and (I) have a suitable fitting effect on the data in the early stage, but the error between predicted values and real values in the late stage is large. Plan (E) has the best fitting effect and the smallest error between real and predicted values.

As can be seen from Table 12, MAE, MSE, and MAPE values of Plan (E) under different hyperparameters are 0.3729%, 0.0024%, and 0.0669%, respectively. It indicates that Plan (E) has the smallest error and the best prediction effect. Therefore, the model parameters with an activation function of the sigmoid, a batch size of 100, and the number of nodes in the network layer of 64 are determined.

**Table 12.** The evaluation criterion of the orthogonal test.

| Plan Number | A | B | C | D | E | F | G | H | I |
|---|---|---|---|---|---|---|---|---|---|
| MSE(%) | 0.0112 | 0.0529 | 0.0159 | 0.0094 | 0.0024 | 0.0382 | 0.0229 | 0.0631 | 0.0356 |
| MAE(%) | 0.8045 | 1.5836 | 0.9254 | 0.7487 | 0.3729 | 1.3125 | 1.1527 | 1.7222 | 1.2970 |
| MAPE(%) | 0.1443 | 0.2840 | 0.1660 | 0.1343 | 0.0669 | 0.2354 | 0.2068 | 0.3088 | 0.2327 |

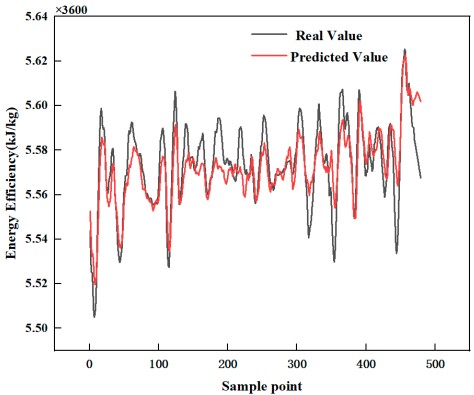

(**A**) Prediction results of plan A

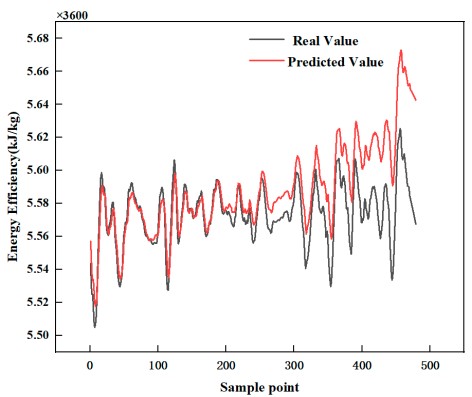

(**B**) Prediction results of plan B

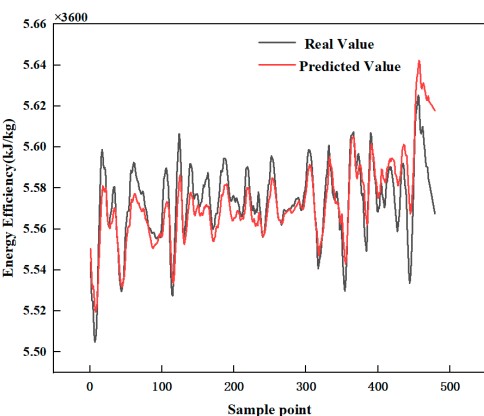

(**C**) Prediction results of plan C

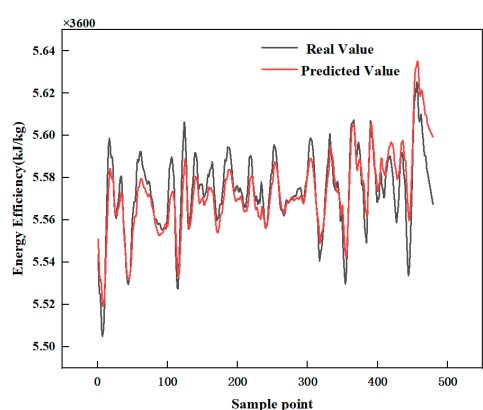

(**D**) Prediction results of plan D

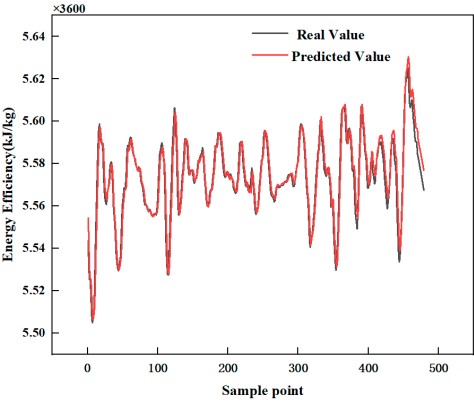

(**E**) Prediction results of plan E

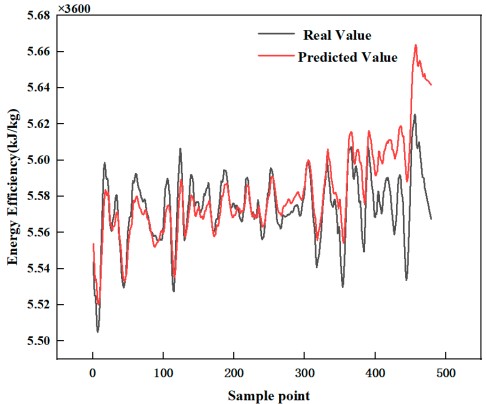

(**F**) Prediction results of plan F

**Figure 11.** *Cont.*

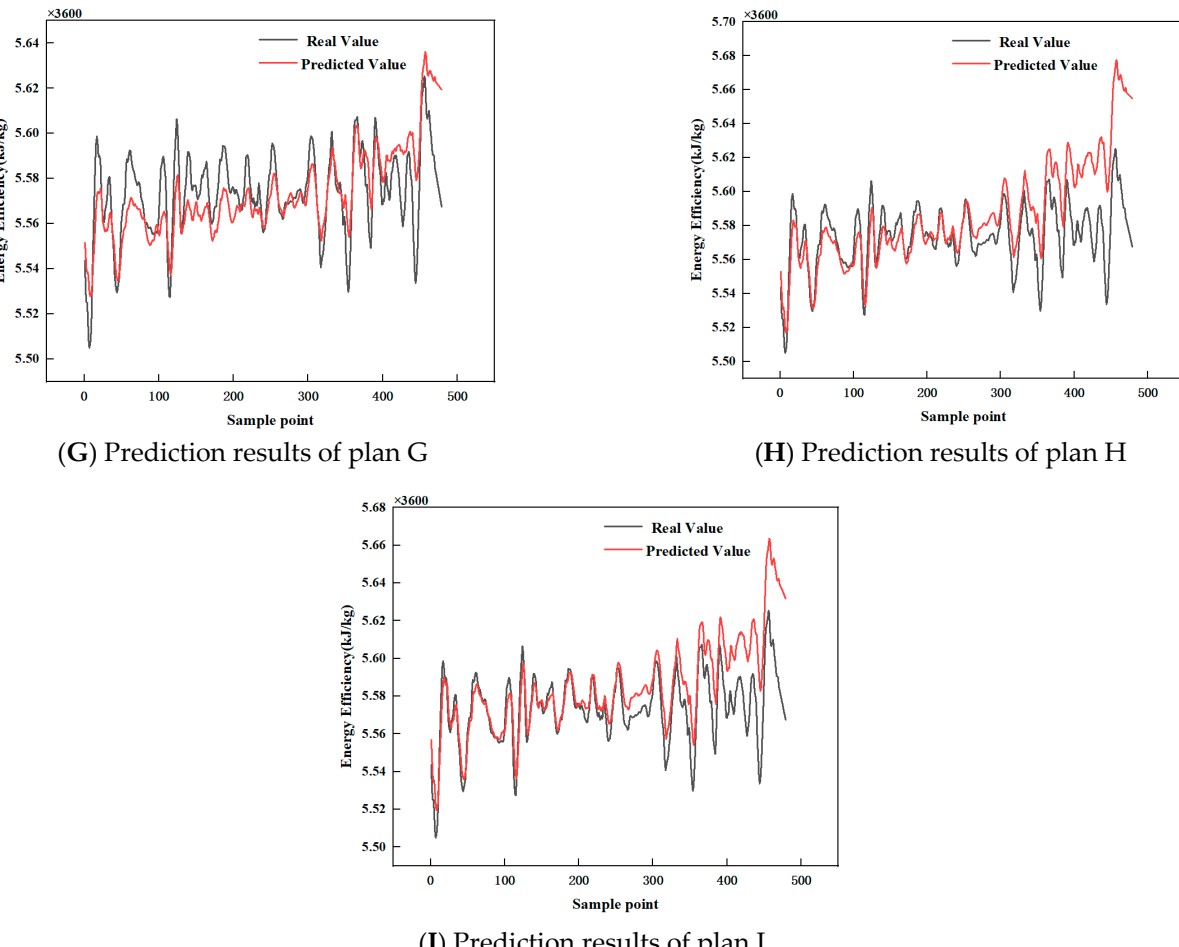

**Figure 11.** Comparison of prediction results of different hyperparameters.

*4.4. Prediction Result Analysis*

The relationship between the operating parameters and energy efficiency is highly coupled and nonlinear. On the basis of establishing the mechanism model, the LSTM model is trained as the agent model. The model can predict the energy efficiency of products in real time according to the changes in operating parameters and estimate the energy-efficiency trend in advance. The hyperparameter-optimized LSTM network is used for prediction, the input parameters of the model are shown in Tables 7 and 8, and the output parameters are the product energy efficiency. Based on the relationship between operating parameters and energy consumption and combined with deep learning, the product energy efficiency can be predicted in real time. By adding disturbances to the operating parameters, it is possible to ensure that the model developed has suitable predictions for different operating conditions.

In the prediction result of LSTM, the horizontal axis is the number of samples in the test set, and the vertical axis indicates the product energy efficiency. The blue line indicates the real energy efficiency of the sideline product, and the red line indicates the predicted energy-efficiency value. To clearly reflect the difference between the predicted value and the real value, subtract the predicted value from the true value, and the difference is shown in the black line. As shown in Figures 12–15, the LSTM model has a high accuracy of energy efficiency under different working conditions and a suitable degree of data fitting, which can obtain relatively accurate energy-efficiency predicted value. This model is of great significance for dynamic monitoring and evaluation of energy efficiency.

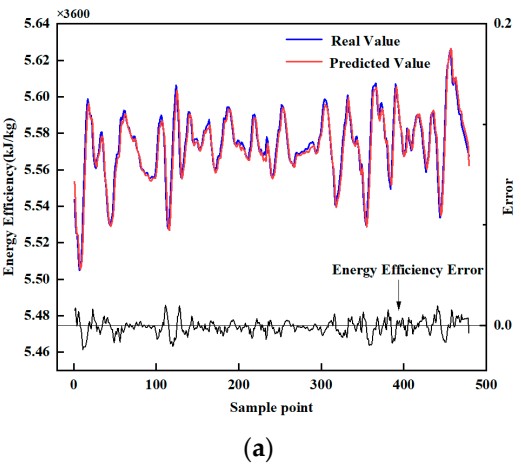
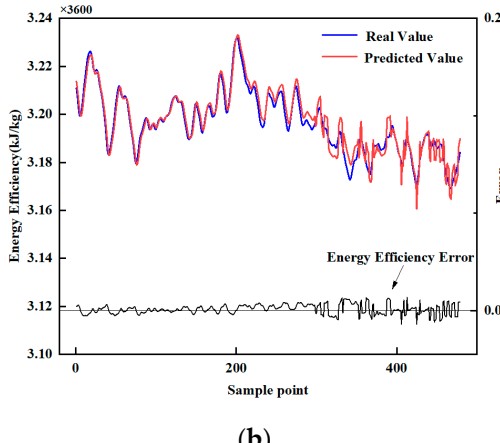

**Figure 12.** Product energy-efficiency prediction of working condition 1. (**a**) Prediction of energy efficiency of atmospheric two-line products. (**b**) Prediction of energy efficiency of vacuum two-line product.

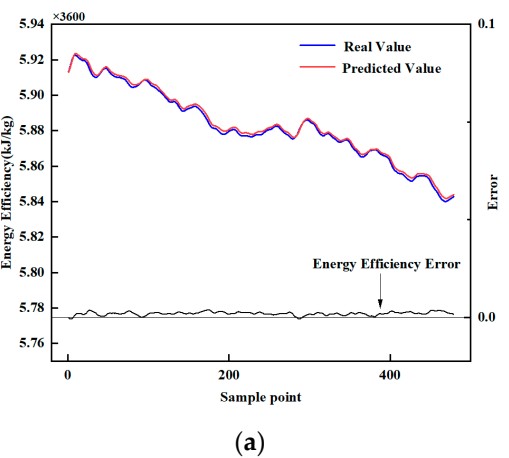
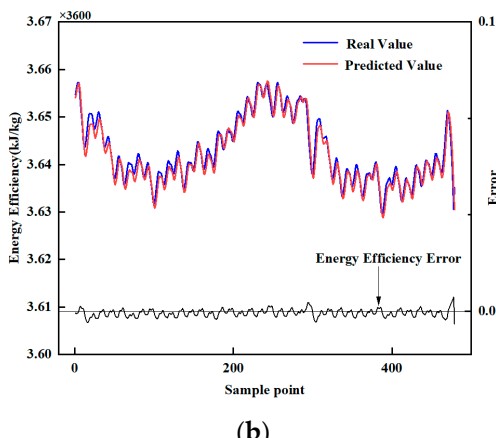

**Figure 13.** Product energy-efficiency prediction of working condition 2. (**a**) Prediction of energy efficiency of atmospheric two-line products. (**b**) Prediction of energy efficiency of vacuum two-line products.

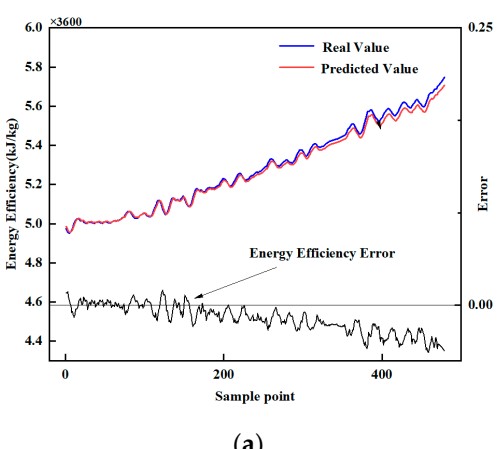
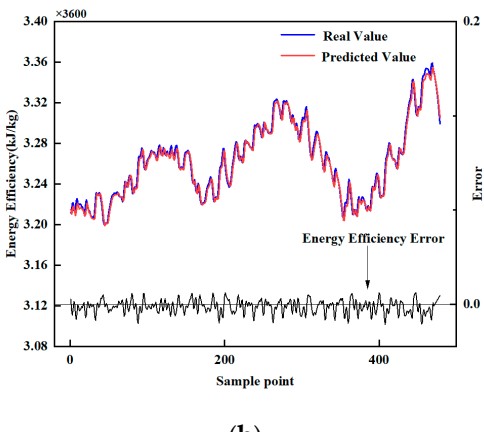

**Figure 14.** Product energy-efficiency prediction of working condition 3. (**a**) Prediction of energy efficiency of atmospheric two-line products. (**b**) Prediction of energy efficiency of vacuum two-line products.

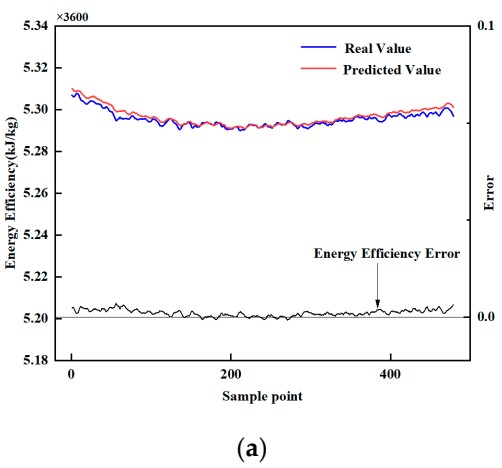 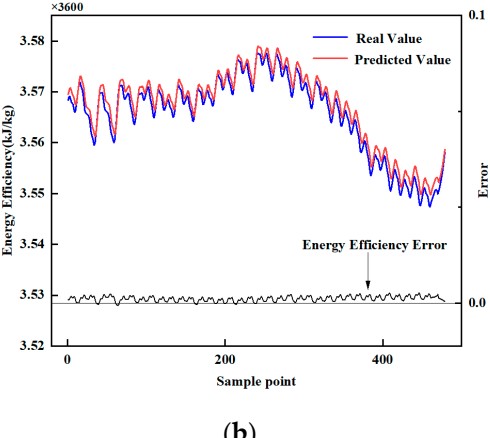

(**a**)  (**b**)

**Figure 15.** Product energy-efficiency prediction of working condition 4. (**a**) Prediction of energy efficiency of atmospheric two-line products. (**b**) Prediction of energy efficiency of vacuum two-line products.

In order to better evaluate the prediction accuracy of the LSTM model, MAPE, MAE, and MSE of the LSTM model under different working conditions are calculated and shown in Table 13. It can be seen that the values of MAE, MSE, and MAPE are lower than 1.3872%, 0.0307%, and 0.2555%, which proves the high accuracy of prediction results of energy efficiency.

**Table 13.** The evaluation criterion of the LSTM model.

| Evaluation Criterion | Condition 1 | | Condition 2 | | Condition 3 | | Condition 4 | |
|---|---|---|---|---|---|---|---|---|
| | (a) | (b) | (a) | (b) | (a) | (b) | (a) | (b) |
| MSE (%) | 0.0024 | 0.0039 | 0.0002 | 0.0002 | 0.0307 | 0.0024 | 0.0003 | 0.0003 |
| MAE (%) | 0.3729 | 0.4635 | 0.1296 | 0.1135 | 1.3872 | 0.3846 | 0.1440 | 0.1501 |
| MAPE (%) | 0.0669 | 0.1453 | 0.0220 | 0.0312 | 0.2555 | 0.1177 | 0.0272 | 0.0421 |

At the same time, the model can also provide a reference for the energy efficiency of the plant. The reference value of energy efficiency can be obtained by bringing the existing data of the plant into the LSTM model. It generates an energy-efficiency diagnostic report for the plant based on the deviation between the reference and actual values. If the actual value of product energy efficiency is lower than the reference value, it can indicate that the energy utilization of the plant is low at that stage. If the actual value of product energy consumption is higher than the reference value of the model or close to the reference value, it can indicate that the energy utilization of the plant is high at that stage. The data where the true value of energy efficiency is greater than the reference value provides a database for improving the energy efficiency of the plant. The operation parameters with high energy efficiency can be used for optimal operation parameters. According to the actual situation of the device and the setting of operating conditions to optimize the operating parameters, the energy efficiency of the unit achieves the optimal state.

## 5. Conclusions

An energy-efficiency prediction method based on mechanism modeling and artificial intelligence is proposed for the crude distillation process to accurately predict and evaluate energy efficiency. The study can be summarized as follows. Firstly, four working conditions of crude distillation are simulated to obtain sample sets for subsequent prediction of energy efficiency. Secondly, the distance-encoded heat map is introduced to visualize specific process parameter changes under abnormal operating conditions to demonstrate the variability in different working conditions. Thirdly, sample sets obtained from each con-

dition are smoothed through SG filters with the aim of reducing data noise while retaining the original features. Finally, the energy-efficiency values predicted by the deep learning LSTM model are compared with real values to assess the reasonableness of the production parameters. The results show that MAE, MSE, and MAPE predicted by the LSTM model are lower than 1.3872%, 0.0307%, and 0.2555% under different working conditions.

This study shows that the LSTM model has high reliability and suitable reference in energy-efficiency prediction in guiding operators to make decisions for operating parameters optimization, energy-efficiency diagnoses, and optimal production strategies design. However, the method proposed belongs to supervised learning. How to carry out unsupervised learning in process data will be the focus of future research. Therefore, we will further study the application of unsupervised learning in energy-efficiency prediction. In addition, we will study and integrate other methods to study the early warning problem of energy efficiency and optimize the operating parameters of the device.

**Author Contributions:** Data curation, investigation, and resources, Y.Z.; methodology and project administration, Z.C.; resources and writing—review, M.W.; supervision and validation, B.L.; visualization and software, X.F.; writing—original draft and funding acquisition, W.T. All authors have read and agreed to the published version of the manuscript.

**Funding:** This research was funded by the National Natural Science Foundation of China (Grant No. 22178189) and the Shandong Natural Science Foundation (Grant No. ZR2021MB113).

**Data Availability Statement:** Not applicable.

**Acknowledgments:** Financial support for carrying out this work is provided by the National Natural Science Foundation of China (grant number: 22178189) and the Shandong Natural Science Foundation (grant no. ZR2021MB113).

**Conflicts of Interest:** The authors declare no conflict of interest.

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
