# Peer review of "An Energy-Efficiency Prediction Method in Crude Distillation Process Based on Long Short-Term Memory Network"

_processes, doi:10.3390/pr11041257_

Round 1
Reviewer 1 Report
Review: processes-2324472
Title: An Energy Efficiency Prediction Method in Crude Distillation 2 Process Based on Long Short-Term Memory Network
The article proposes a model based on a long short-term memory network (LSTM) to predict and analyze energy efficiency in a crude distillation process. I understand that this study is interesting and agrees with the research published by the Processes MDPI. Nevertheless, some adjustments must be made before this occurs. The most important points from this review are described below.
Abstract:
The abstract describes the purpose of the study and the steps taken in its development. However, results are not described in that section, which would be desirable, especially in quantitative terms. As a result, the conclusions reached by the authors are not adequately funded.
Introduction:
The Introduction section is well structured and has been worded accurately. It was possible to observe that the recent major studies in the area are cited in that literature review.
However, what gap this research intends to fill needs to be clarified. Is it just a variation of other studies already done? This aspect needs to be better defined.
In addition, it would be convenient that before the executive summary that closes the section, the authors indicate who would be the direct and indirect beneficiaries of the findings obtained by the research.
Methodology and theoretical basis
Line 122: Although it is a common practice to simulate processes (or operations) in a steady state, this condition is particular and only sometimes frequent. Therefore, the authors must justify such a simplification of the model and explain what consequences this decision can bring to the results.
Conclusions: I missed a passage describing the study’s limitations and other problems that might affect the results obtained and increase their uncertainties. Please comment on such points.
Line 128: How were disadvantages provided by the Savitzky-Golay filters (e.g., suppression of high frequencies and artifacts when using polynomial fits for the first and last points) overcome in this case? Did the choice of this digital filter influence the research findings in any way? Please, justify.
Lines 134-147: The authors describe features of the Aspen Hysys software, among which is the possibility of dynamic simulation. As far as I can note, this feature has not been used. (Please see previous comment on the topic). Therefore, I suggest reviewing the content of section 2.2 of the manuscript (possibly removing this content and adding other aspects of the tool that helped develop the study) to prevent the reader from having an inadequate understanding of how the research was conducted.
In the version of the manuscript I accessed, it is impossible to know (and from that, analyze) the content of the equations described in lines 172-174 and Line 190.
The abovementioned problem also occurs with Antoine’s equation (Line 233). Indeed, this model even relates temperature and saturation pressure very simply. I mean, other approaches are more consistent and rigorous than this one. So why was it chosen?
The parameters in Table 1-5 correspond to a possible operational situation for atmospheric distillation. I saw no indication in the text that this would be the most likely (or frequent) condition of using the equipment. Does this happen? Why were operating ranges not evaluated? Using this approach would reduce the deterministic character of the model and, thus, expand the spectrum (entirely restricted at this moment) of conclusions reached by the research.
Line 309: equation n. 7 also have expression problems. The same occurs with Equations 8 (Line 392) and 9 (Line 398). These limitations greatly marred my review. It is impossible to know (and sometimes even intuit) the approach given by the authors in each situation or stage of the methodology.
Figures 6-9 (a – b): It would be fundamental for the understanding of the analysis that the resolution of these figures was improved. This aspect is essential.
Conclusions:
The Conclusions section deviates from the structure conventionally adopted for its writing. Despite this, I understand that the research generated many findings that should have been highlighted at this moment in the manuscript. I would suggest you review the content to make any necessary improvements.
Reviewer 2 Report
This paper proposes a predictive analysis model based on long and short-term memory network (LSTM), and applies it to the energy efficiency cost optimization problem of crude distillation process in petrochemical industry. The research results have certain theoretical significance and practical value, and the technical route is clear and innovative.
Specific suggestions are as follows:
1. Please further elaborate on the scope of application and improvement methods of long and short-term memory network model in Part 2.
2.3.3 Unreasonable content setting, short length and poor support for the article.
3. Please elaborate on the simulation process and results in various scenarios in Section 4, and add other scenarios to demonstrate the applicability of the method.
4. Please give a more detailed and convincing explanation for the selection of original data in the prediction model experiment in Part 4;
5. It is suggested to adjust the layout. Some pictures lack units and some tables are not uniform in format, so they need to be modified.
6. In Section 5, please add the guiding significance and prospects of the conclusions for practical problems.
7. Some references were published earlier, so it is suggested to update them. For example, references [1], [2] and [3] were published earlier than 2014.
Reviewer 3 Report
This paper provides a Prediction Method in Crude Distillation. Although the topic is interesting and the quality of the paper is acceptable, there are some points that deserve to be revised.
It is suggested to re-design figure 1.
I suggest describing the motivations and the objectives of the proposed research clearly and concisely.
I recommend explaining the axis values of Figures 10. All figures seem to be similar. I would express results more accurately in a single figure. In addition to MAPE, I suggest evaluating by metrics such as MAE or MSE
The equations are not displayed correctly. It may be a problem with the template.
RNNs are suitable for time-series and sequential data, the order of input matters is important as RNN proposal can model time dependencies. Why authors have they decided to use this method? And particularly LSTM? It should be clearly discussed.
Description of figure5 and tables are suggested at the bottom.
Please list more about the limitation, I am not very clear about it.
Round 2
Reviewer 1 Report
Dear Authors,
an analysis of the revised version (v2) of manuscript processes-2324472, indicated that most of the suggestions for correction, adequacy, and complementation that I had made on the occasion of the first round of revisions of that content were carried out. Even though I can still see aspects to be improved in the same document, its current form perfectly meets the expectations of a publication carried out by Processes - MDPI. My recommendation is, therefore, that it be communicated in this format. Congratulations.
Reviewer 3 Report
Authors addressed all the comments and suggestions made. For instance, Authors have redesigned the flow chart to show the research process and goal more directly. Authoes have also included MAE and MSE evaluation indexes.
I believe that the manuscript can be processed for publication in its current form.